# Predictive and Diagnostic Values of Systemic Inflammatory Indices in Bronchopulmonary Dysplasia

**DOI:** 10.3390/children11010024

**Published:** 2023-12-25

**Authors:** Linxia Cao, Xiangye Liu, Tingting Sun, Yuan Zhang, Tianping Bao, Huaiping Cheng, Zhaofang Tian

**Affiliations:** Department of Neonatology, The Affiliated Huaian No.1 People’s Hospital of Nanjing Medical University, Huai’an 223300, China; linxia_cao@163.com (L.C.); liuxiangye1990@126.com (X.L.); stt201508@126.com (T.S.); zhangyuan8505@126.com (Y.Z.); hayybtp@njmu.edu.cn (T.B.); hayy1316@163.com (H.C.)

**Keywords:** systemic inflammation indices, bronchopulmonary dysplasia, preterm infants

## Abstract

Background: Bronchopulmonary dysplasia (BPD) is the most common respiratory complication in preterm infants, and there is a lag in the diagnosis of BPD. Inflammation is a vital pathogenic factor for BPD; we aim to evaluate the predictive and diagnostic values of systemic inflammatory indices in BPD. Methods: Between 1 January 2019 and 31 May 2023, the clinical data of 122 premature infants with a gestational age of <32 weeks in the Department of Neonatology, the Affiliated Huai’an No. 1 People’s Hospital of Nanjing Medical University, were retrospectively collected and classified into non-BPD (*n* = 72) and BPD (*n* = 50) groups based on the National Institute of Child Health and Human Development 2018 criteria. To compare the general characteristics of each group, we identified the independent risk variables for BPD using multivariate logistic regression analysis, compared the systemic inflammatory indices at birth, 72 h, 1 week, 2 weeks, and 36 weeks postmenstrual age (PMA), and constructed the receiver operating characteristic curves of neutrophil-to-lymphocyte ratio (NLR) diagnosis of BPD at different time points. Results: ① The independent risk factors for BPD in preterm infants were birth weight, small for gestational age, and days of oxygen therapy (all *p* < 0.05). ② At 72 h and 1 week after birth, the serum NLR of the BPD group was higher than for the non-BPD group (*p* < 0.05). Furthermore, the neutrophil count (N), NLR, monocyte-to-lymphocyte ratio (MLR), systemic immune-inflammation index, systemic inflammation response index (SIRI), and pan-immune-inflammation value of infants with BPD were higher than the non-BPD group at 3 weeks after birth (*p* < 0.05). Moreover, at 36 weeks of PMA, the serum N, NLR, MLR, and SIRI of BPD infants were higher than those of non-BPD infants (*p* < 0.05). ③ The NLR of infants with and without BPD gradually increased after birth, reaching a peak at 72 h and 1 week, respectively. At 3 weeks postnatal, the NLR had the highest predictive power for BPD, with an area under the curve (AUC) of 0.717 (*p* < 0.001); the sensitivity was 56% and specificity was 86.1%. In addition, the NLR at 36 weeks of PMA exhibited some diagnostic value for BPD. The AUC was 0.693 (*p* < 0.001), the sensitivity was 54%, and specificity was 83.3%. Conclusions: At 3 weeks after birth and 36 weeks of PMA, some systemic inflammation indices (like N, NLR, SIRI) of preterm infants with BPD have specific predictive and diagnostic values; these indices may help the management of high-risk preterm infants with BPD.

## 1. Introduction

Recently, numerous premature infants have survived because of the rapid advances in healthcare. Nevertheless, the prevalence of bronchopulmonary dysplasia (BPD) has increased. From 2009 to 2012, the incidence of BPD increased at all gestational ages except for 28 weeks [1]. The incidence of BPD in premature infants with a gestational age < 32 weeks in Suzhou from 2016 to 2021 was 25.36% [2]. BPD is a multifactorial disease [3] caused by the interaction between genetic and environmental factors. At present, it is hypothesized that BPD occurs because of a combination of inflammation, injury to immature lung development, and impaired growth of the alveoli and blood vessels [4]. The manifestations of BPD include lung growth arrest, alveolar simplification, decreased vascular development, and aberrant lung function; these manifestations are closely linked to recurrent hospitalization, neurodevelopmental difficulties, and chronic lung illnesses [5]. Inflammation is a vital pathogenic factor for BPD and there is a lag in the diagnosis of BPD; therefore, it is crucial to predict BPD occurrence by studying some inflammatory markers; research into these predictive markers may help us to intervene in high-risk patients with BPD as early as possible.

Some hematological indicators of inflammation, including the neutrophil-to-lymphocyte ratio (NLR) and platelet-to-lymphocyte ratio (PLR), can be considered as novel potential markers for evaluating systemic inflammation; these indicators are closely associated with the clinical prognosis of several respiratory diseases, including chronic obstructive pulmonary disease (COPD) [6], COVID-19 [7], idiopathic pulmonary fibrosis [8] and non-small cell lung cancer [9]. These indicators are inexpensive, effective, and convenient.

Recent studies have reported that some inflammatory indicators are closely associated with BPD [10,11,12]; however, these inflammatory markers for BPD have unknown predictive and diagnostic values. Because inflammation has a significant impact on the onset and progression of BPD, we sought to assess the predictive and diagnostic values of serum systemic inflammatory indices such as the neutrophil count (N), NLR, monocyte-to-lymphocyte ratio (MLR), PLR, pan-immune-inflammation value (PIV), systemic immune-inflammation index (SII) and systemic inflammation response index (SIRI) for BPD at birth, 72 h, 1 week, 2 weeks, 3 weeks and 36 weeks of postmenstrual age (PMA).

## 2. Material and Methods

### 2.1. Study Subjects

This study was carried out in the Department of Neonatology, the Affiliated Huai’an No. 1 People’s Hospital of Nanjing Medical University, between Jan 1, 2019 and May 31, 2023. The essential information and clinical data were obtained from the electronic medical record system of the hospital, and 188 patients with a gestational age of <32 weeks were enrolled. Among the enrolled patients, 46 with a length of hospital stay of <21 days and 20 who were discharged before 36 weeks of postmenstrual age (PMA) were excluded from the study. Finally, the study included 122 eligible premature infants. According to the 2018 National Institute of Child Health and Human Development (NICHD) criteria [13], the infants were classified into two groups: non-BPD (72 cases) and BPD (50 cases). Furthermore, based on the severity of the BPD, BPD Grade I (23 cases), BPD Grade II (12 cases), and BPD Grade III (15 cases) were the three categories into which the patients were divided. The following were the inclusion requirements: ① if the gestational age was <32 weeks, infants were transferred to our NICU within 24 h of delivery, and ② they remained hospitalized beyond 36 weeks of PMA. The following were the exclusion requirements: ① infants with severe congenital diseases, congenital malformations, diaphragmatic hernia, genetic metabolic diseases, etc., and ② those who were discharged before reaching 36 weeks of PMA and those with a length of hospital stay of <21 days. This study was endorsed by the Ethics Committee of the Affiliated Huai’an No. 1 People’s Hospital of Nanjing Medical University (Ethical approval No.: KY-2023-118-01 and approved on Jul 19, 2023). Family members of the newborns gave their written consent to participate in the study and study approval.

### 2.2. Outcome Measures

The following characteristics of infants with and without BPD were compared: gestational age; birth weight; the number of fetuses; delivery mode; Apgar score; gender; small for gestational age children (SGA); postpartum dexamethasone treatment; exogenous surfactant drug administration; length of hospital stay; total time of O_2_ therapy; 5 L/min of O_2_ inhalation in the box; 5 or 7 L/min of O_2_ inhalation in the hood; continuous positive airway pressure (CPAP) time; mechanical ventilation time; the presence of complications such as the premature rupture of membranes, newborn respiratory distress syndrome, early-onset sepsis, congenital pneumonia, patent ductus arteriosus, pulmonary hypertension, prematurity-related retinopathy, neonatal intracranial hemorrhage, neonatal necrotizing enterocolitis; maternal age and maternal health such as pregnancy-induced hypertension and pregnancy-induced diabetes. The fifth edition of *Practice of Neonatology* served as the foundation for the diagnostic criteria for complications [14].

All infants were given peripheral venous blood samples at birth, 72 h after birth, 1 week, 2 weeks, 3 weeks and 36 weeks of PMA. Blood was collected into ethylenediaminetetraacetic acid tubes. Complete blood count analysis was performed using SYSMEX XN-9000 fully automatic blood cell line equipment (KHB, Shanghai). The N (10^9^/L), platelet count (10^9^/L) (P), monocyte count (10^9^/L) (M), and lymphocyte count (10^9^/L) (L) were recorded. Furthermore, the systemic inflammatory indices (N, NLR = N/L, PLR = P/L, MLR = M/L and SII = N × P/L, SIRI = N × M/L and PIV = P × N × M/L.) of each group of preterm infants at birth, 72 h after birth, 1 week, 2 weeks, 3 weeks and 36 weeks of PMA were compared.

### 2.3. Statistical Analysis

When the measurement data were normally distributed, comparing the two groups, we used the t (t’) test; results were presented as mean ± standard deviation ($\bar{\chi }$ ± s). The median (quartile) [M (P25, P75)] was used to represent data that were not regularly distributed. For the comparison of the two groups we used the Wilcoxon test, whereas for the comparison of multiple groups we used the Kruskal–Wallis test, and multiple comparisons were conducted. Counting data were converted to percentages (%) using either the Pearson’s χ2 test or the Fisher’s exact test. Furthermore, the risk variables affecting BPD were found using multivariate logistic regression analysis, adjusted for birth weight, SGA, days of O2 therapy, O2 inhalation in the box (5 L/min), days of CPAP therapy, mechanical ventilation time and N, NLR, SIRI at 3 weeks. The receiver operating characteristic curve (ROC) of the subjects was constructed, and the area under the curve (AUC) values were obtained. A *p*-value of <0.05 was considered to indicate statistically significant differences. A SPSS 25.0 version was used to examine data.

## 3. Results

### 3.1. Baseline Characteristics of Infants with and without BPD

Birth weight, multiple pregnancies, SGA, postpartum dexamethasone treatment, exogenous surfactant administration, length of hospital stay, total time of O_2_ therapy, time of 5 L/min of O_2_ inhalation in the box, CPAP time, mechanical ventilation time, and early-onset sepsis were identified as the pertinent risk factors for BPD (*p* < 0.05, Table 1).

### 3.2. Multivariate Logistic Regression Analysis

To characterize independent variables that are associated with BPD, we used multivariate analysis adjusted for birth weight, SGA, days of O2 therapy, O2 inhalation in the box (5 L/min), days of CPAP therapy, mechanical ventilation time and N, NLR, and SIRI at 3 weeks. As shown in Table 2, the effects of birth weight (OR = 1.004, 95% CI: 1.000–1.008, *p* < 0.05), SGA (OR = 0.021, 95% CI: 0.001–0.323, *p* < 0.05), and days of O2 therapy (OR = 1.189, 95% CI: 1.004–1.407, *p* < 0.05) on BPD were statistically significant.

### 3.3. Comparison of the Systemic Inflammatory Indices of the Two Premature Infant Groups

There were no statistically significant differences in systemic inflammatory indices between the BPD and non-BPD groups at birth and 2 weeks after birth; however, the NLR was higher in the BPD groups than in the non-BPD groups at 72 h and 1 week after birth. Other inflammatory indices showed no statistically significant differences (see Appendix A).

At 3 weeks, the systemic inflammatory indices of infants with BPD were compared between the two groups. N, NLR, MLR, SII, SIRI, and PIV were higher in the BPD group than in the non-BPD group (*p* < 0.05, Table 3). However, there was no statistically significant difference in the PLR.

At 36 weeks of PMA, the N, NLR, MLR, and SIRI of infants with BPD were higher than those without BPD (*p* < 0.05, Table 4). However, there were no statistically significant differences in the PLR, SII, and PIV between the two groups. 

### 3.4. Value of NLR in BPD

#### Dynamic Changes in NLR

The serum NLR of infants with and without BPD gradually increased after birth, reaching a peak at 72 h and 1 week, respectively, and decreasing after that (Figure 1).

### 3.5. ROC Analysis of NLR to Predict BPD at Different Time Points

We compared the ROC curves of the NLR of preterm infants with BPD at different time points (Figure 2, Table 5) and observed that the NLR had the highest predictive power for BPD at 3 weeks after birth, with an AUC of 0.717 (*p* < 0.001) and 95% CI of 0.621–0.813. The maximum value of Youden’s J statistic was 0.421; furthermore, the optimal cutoff value at this time was 1.416, the sensitivity was 56%, and the specificity was 86.1%.

### 3.6. ROC Analysis of NLR for BPD Diagnosis

For the ROC curves (Figure 3) of preterm infants diagnosed with BPD using the NLR at 36 weeks of PMA, the AUC was 0.693 (*p* < 0.001), and 95% CI was 0.595–0.792. The maximum value of Youden’s J statistic was 0.373 and the optimal cutoff value was 1.1125 at this time; the sensitivity was 54% and the specificity was 83.3%.

## 4. Discussion

BPD is a multifactorial illness that occurs in premature infants who receive mechanical ventilation and O_2_ supplementation, ultimately resulting in long-term lung disease. At present, BPD is no longer considered a neonatal disease but rather a lifelong disease [15]. Therefore, it is vital to simply and effectively detect BPD for early diagnosis and treatment.

In our study, we used the 2018 NICHD diagnostic criteria to conduct a univariate analysis of the clinical data of two groups. We observed that birth weight, multiple births, small for gestational age, postpartum dexamethasone treatment, exogenous surfactant administration, length of hospital stay, days of O_2_ therapy, time of 5 L/min of O_2_ inhalation in the box, CPAP time, mechanical ventilation time, and early-onset sepsis were higher in the BPD group than in the non-BPD group; these findings are consistent with those of Sun et al. [16]. Furthermore, multivariate logistic regression analysis revealed that birth weight, SGA, and days of O2 therapy were independent risk factors for BPD. Some studies [17] have hypothesized that birth weight is the primary endogenous risk factor for inflammation and that inflammation is an essential factor causing BPD; therefore, low-birth-weight children are more likely to suffer from BPD. Another study [18] has reported that SGA is a pathogenic factor for BPD; this finding is consistent with that of the present study and may be related to the impairment of the alveolar formation and imbalance in cytokine response in small-for-gestational-age infants. O_2_ toxicity is one of the reasons for BPD development [13].

Systemic inflammatory indices are crucial in many respiratory diseases. For example, the NLR and PLR of patients with COPD increase during the stable phase and further increase during the exacerbation phase; this can help predict hospitalization mortality [19]. Inpatients with an acute exacerbation of COPD within 90 days of discharge are at a higher risk of COPD-related mortality, which is why the NLR is crucial for predicting this risk [20]. Furthermore, the NLR is a valid marker for predicting the prognosis of patients with COVID-19, and elevated levels of NLR and SII play vital roles in predicting the prognosis of patients with severe COVID-19 [21,22]. Increased levels of NLR, MLR, and SII are precious for predicting COVID-19-related mortality [23,24]. Based on the severity of hypoxia, there may be changes in the degree of hyperoxia and inflammation in children with BPD and the number and function of neutrophils, monocytes, and lymphocytes [25]. The systemic inflammatory state and immune response can be reflected by inflammatory biomarkers such as the NLR, SII, and SIRI [26]. Physiological, pathological, and physical factors minimally affect these fast and convenient biomarkers.

We compared and analyzed the systemic inflammatory indices of infants with and without BPD at different time points. The N, NLR, MLR, SII, SIRI, and PIV of BPD groups were higher than those of non-BPD groups at 3 weeks; this finding suggests that these indices predict BPD occurrence. Furthermore, at 36 weeks of PMA, the N, NLR, MLR, and SIRI of BPD infants were higher than those of non-BPD infants; this indicates that these indices have some diagnostic value for BPD. Showing the online dynamic link between innate (neutrophils) and adaptive (lymphocytes) immune responses during various illnesses and pathological states, NLR is a developing marker of cellular immune activation and an efficient indication of stress and systemic inflammation [27]. We observed that the serum NLR of BPD infants was higher than that of non-BPD infants at 72 h, 1 week, 3 weeks and 36 weeks of PMA. Furthermore, the NLR of infants with and without BPD gradually increased after birth, reaching a peak at 72 h and 1 week, respectively, and then decreased; this is consistent with the inflammatory effect of BPD. However, the NLR of the two groups showed no significant difference at birth and 2 weeks. Sun et al. [16] reported that the serum NLR of infants with BPD at birth and 72 h was higher than that of those without BPD and that the two groups showed no significant difference at 1 week and 2 weeks. These findings may be related to the low percentage of OLFM4^+^ neutrophils, a subset of neutrophils, in children with BPD at birth [28]; furthermore, we adopted the diagnostic criteria of 2018, whereas Sun et al. used the BPD diagnostic criteria of 2001. These two are different in the inclusion population, assessment nodes, and classification. Cakir et al. [10] reported systemic inflammatory indices in bronchopulmonary dysplasia at birth and 36 weeks of PMA, and the results showed a difference only with SIRI, which is inconsistent with our findings. It may be due to the different diagnostic criteria and grouping methods we employ. Notably, under physiological conditions, the number of neutrophils and lymphocytes is equal at 4–6 days after birth; therefore, an increase in the serum NLR of infants with BPD at 72 h and 1 week after birth has a higher predictive value. In addition, we compared the ROC curves of the serum NLR of preterm infants at different time points with BPD occurrence. The NLR had the highest predictive power for BPD at 3 weeks after birth, with an AUC of 0.717, an optimal cutoff value of 1.416, a sensitivity of 56%, and a specificity of 86.1%. At 36 weeks of PMA, the AUC of NLR for BPD diagnosis in premature infants was 0.693, with an optimal cutoff value of 1.1125, a sensitivity of 54%, and a specificity of 83.3%. To sum up, these findings suggest that the NLR can predict BPD occurrence at 3 weeks after birth and help diagnose BPD at 36 weeks of PMA, with some level of specificity.

In our study, the incidence of EOS in the BPD group was higher than that in the non-BPD group, which may be related to the fact that EOS can be used as a trigger of BPD. During the study period, we administered low-dose dexamethasone for high-risk infants of BPD (postnatal mechanical ventilation for more than a week), which led to an increased incidence of glucocorticoid use in the BPD group. It may have been associated with the elevation of these indicators. The influence of the above two aspects on the study’s results needs to be considered in further multicenter, large-scale clinical research.

In conclusion, we observed that some systemic inflammatory indices (like N, NLR, and SIRI) have specific clinical value in predicting and diagnosing BPD at 3 weeks after birth and 36 weeks of PMA, particularly the NLR, which has the highest predictive value. The limitation of our study includes the small number of EOS cases and the greater use of postnatal steroids. This paper is a single-center clinical study with a small sample size. There is also a lack of research on the correlation between these inflammatory indices and the practical measures for BPD. Further studies on these inflammatory markers will provide new ideas for the clinical management of BPD.

## Figures and Tables

**Figure 1 children-11-00024-f001:**
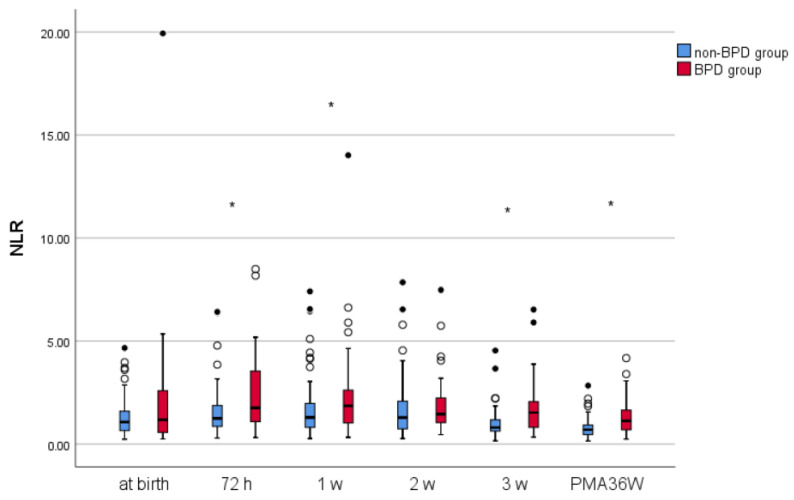
Box plot of serum neutrophil-lymphocyte ratio (NLR) at different time points. Black and white circle means outliers. * *p* < 0.05.

**Figure 2 children-11-00024-f002:**
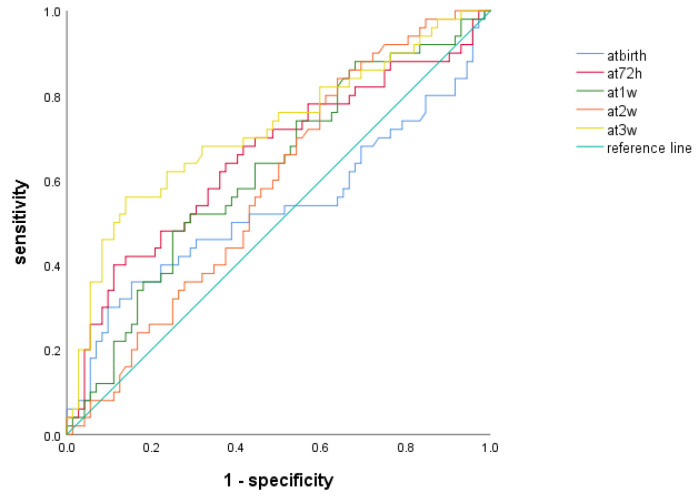
Receiver operating characteristic curve of NLR for BPD development at different time points.

**Figure 3 children-11-00024-f003:**
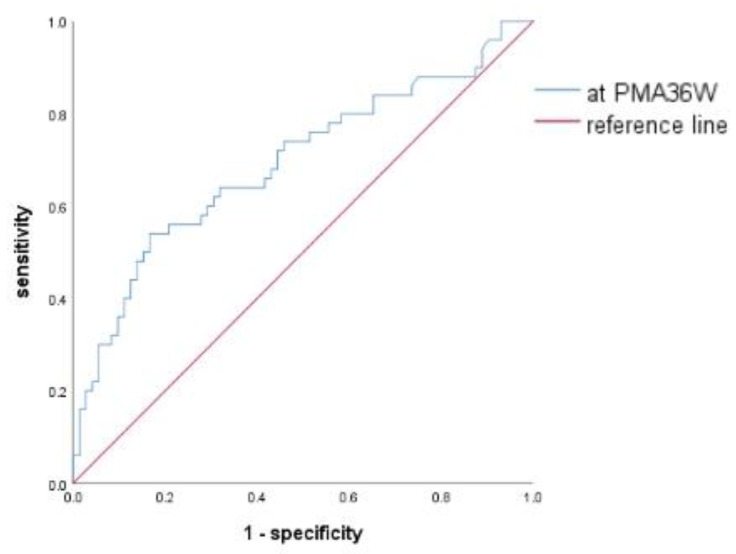
Receiver operating characteristic curve of NLR for BPD development at 36 weeks of postmenstrual age.

**Table 1 children-11-00024-t001:** Baseline characteristics of preterm infants with and without BPD.

	Non-BPD (n = 72)	BPD (n = 50)	*p*
Gestational age (weeks) ^a^	30.14 (29.43, 31.25)	30.29 (28.93, 31.18)	0.739
Birth weight (g) ^b^	1389.44 ± 220.56	1251 ± 294.16	0.006 *
Multiple pregnancies, n (%)	13 (18.1)	17 (34)	0.044 *
Cesarean delivery, n (%)	28 (38.9)	28 (56)	0.062
Apgar score at 1 min < 7, n (%)	10 (13.9)	8 (16)	0.746
Apgar score at 5 min < 7, n (%)	3 (4.2)	6 (12)	0.202
Male, n (%)	27 (37.5)	24 (48)	0.248
SGA, n (%)	5 (6.9)	15 (30)	0.001 *
Postnatal dexamethasone treatment, n (%)	21 (29.2)	38 (76)	0.000 *
Surfactant administration, n (%)	50 (69.4)	43 (86)	0.035 *
Duration of hospitalization (days) ^a^	48.5 (41.25, 56)	65.5 (53.75, 77.5)	0.000 *
Days of O_2_ therapy (days) ^a^	22.88 (13.00, 32.05)	49.66 (37.38, 59.62)	0.000 *
O_2_ inhalation in the box (5 L/min)(days) ^a^	3.01 (1.94, 7.09)	7.26 (4.52, 11.11)	0.001 *
Hood O_2_ (5 or 7 L/min)(days) ^a^	2.04 (0, 4.99)	1.98 (0.42, 9.50)	0.242
Days of CPAP therapy (days) ^a^	7.34 (4.39, 16.63)	14.10 (6.97, 26.47)	0.001 *
Mechanical ventilation time (days) ^a^	0 (0,6.76)	7.72 (0, 29.28)	0.000 *
PROM, n (%)	33 (45.8)	18 (36)	0.279
NRDS, n (%)	63 (87.5)	48 (96)	0.197
EOS, n (%)	2 (2.8)	13 (26)	0.000 *
Congenital pneumonia, n (%)	5 (6.9)	6 (12)	0.524
PDA, n (%)	24 (33.3)	19 (38)	0.596
PH, n (%)	18 (25)	12 (24)	0.900
ROP, n (%)	22 (30.6)	17 (34)	0.688
Neonatal intracranial hemorrhage, n (%)	57 (79.2)	42 (84)	0.502
NEC, n (%)	4 (5.6)	4 (8)	0.869
Maternal age (year) ^b^	28.76 ± 4.52	30.38 ± 5.19	0.070
Gestational hypertension, n (%)	17 (23.6)	19 (38)	0.087
Gestational diabetes mellitus, n (%)	17 (23.6)	5 (10)	0.054

BPD, bronchopulmonary dysplasia; SGA, small for gestational age; CPAP, continuous positive airway pressure; PROM, premature rupture of membranes; NRDS, newborn respiratory distress syndrome; EOS, early-onset sepsis; PDA, patent ductus arteriosus; PH, pulmonary arterial hypertension; ROP, retinopathy of prematurity; NEC, necrotizing enterocolitis. ^a^ Median (quartile) [M (P25, P75)], ^b^ Mean ± standard deviation. * *p* < 0.05.

**Table 2 children-11-00024-t002:** Multivariate logistic regression analysis.

Influencing Factors	B	SE	Wald X2	OR	95% CI	*p*
Birth weight	0.004	0.002	5.029	1.004	1.000–1.008	0.025 *
SGA	−3.878	1.403	7.644	0.021	0.001–0.323	0.021 *
Days of O2 therapy	0.173	0.086	4.042	1.189	1.004–1.407	0.044 *

SGA, small for gestational age; OR, odds ratio; CI, confidence interval. * *p* < 0.05.

**Table 3 children-11-00024-t003:** Comparison of the systemic inflammatory indices of infants with and without BPD at 3 weeks.

3 Weeks	Non-BPD (n = 72)	BPD (n = 50)	*p*
N^a^	4.44 (3.27, 6.12)	5.97 (4.16, 9.72)	0.001 *
NLR^a^	0.80 (0.62, 1.81)	1.53 (0.79, 2.06)	0.000 *
PLR^a^	55.06 (41.02, 72.99)	48.45 (31.60, 71.21)	0.200
MLR^a^	0.27 (0.22, 0.37)	0.35 (0.25, 0.53)	0.026 *
SII^a^	256.84 (141.69, 394.04)	344.77 (199.98, 595.75)	0.035 *
SIRI^a^	1.28 (0.70, 1.89)	2.29 (1.12, 4.05)	0.001 *
PIV^a^	355.99 (180.40, 664.01)	686.03 (190.20, 1315.89)	0.033 *

N, neutrophil count; NLR, neutrophil-to-lymphocyte ratio; PLR, platelet-to-lymphocyte ratio; MLR, monocyte-to-lymphocyte ratio; SII, systemic immune-inflammation index; SIRI, systemic inflammation response index; PIV, pan-immune-inflammation value. ^a^ Median (quartile) [M (P25, P75)]. * *p* < 0.05.

**Table 4 children-11-00024-t004:** Systemic inflammatory indices of the infants with and without BPD at 36 weeks of postmenstrual age.

PMA 36 Weeks	Non-BPD (n = 72)	BPD (n = 50)	*p*
N^a^	3.50 (2.63, 5.05)	5.09 (3.05, 8.54)	0.003 *
NLR^a^	0.70 (0.46, 0.92)	1.12 (0.68, 1.65)	0.000 *
PLR^a^	64.43 (52.40, 88.02)	62.81 (38.54, 77.33)	0.138
MLR^a^	0.25 (0.18, 0.33)	0.30 (0.23, 0.45)	0.022 *
SII^a^	253.78 (160.09, 348.50)	276.60 (140.87, 609.69)	0.338
SIRI^a^	0.90 (0.55, 1.46)	1.62 (0.89, 3.82)	0.002 *
PIV^a^	332.14 (196.83, 529.39)	412.80 (197.86, 1311.62)	0.252

N, neutrophil count; NLR, neutrophil-to-lymphocyte ratio; PLR, platelet-to-lymphocyte ratio; MLR, monocyte-to-lymphocyte ratio; SII, systemic immune-inflammation index; SIRI, systemic inflammation response index; PIV, pan-immune-inflammation value. ^a^ Median (quartile) [M (P25, P75)]. * *p* < 0.05.

**Table 5 children-11-00024-t005:** Receiver operating characteristic curve of NLR for BPD development at different time points. * *p* < 0.05.

	AUC	95%CI	Cutoff Level	Sensitivity (%)	Specificity (%)	Youden Index	*p*
At birth	0.534	0.422–0.645	1.914	36	84.7	0.207	0.525
At 72 h	0.653	0.551–0.755	2.429	40	88.9	0.289	0.004 *
At 1 w	0.620	0.519–0.721	1.911	48	75	0.23	0.025 *
At 2 w	0.584	0.484–0.685	0.8735	84	36.1	0.201	0.114
At 3 w	0.717	0.621–0.813	1.416	56	86.1	0.421	0.000 *

## Data Availability

The dataset generated or analyzed during this study can be made available to interested researchers by the authors of his article upon reasonable request.

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
