# Peer review of "Predictive and Diagnostic Values of Systemic Inflammatory Indices in Bronchopulmonary Dysplasia"

_children, 2023, doi:10.3390/children11010024_

Round 1

Reviewer 1 Report (Previous Reviewer 2)

Comments and Suggestions for Authors

Thank you for incorporating the changes previously recommended in the manuscript.

Few minor corrections:

1. Conclusion:  At 3 weeks after birth and 36 weeks of PMA, some systemic inflammation indi-ces (like NNLRSIRI) of preterm infants with BPD have certain predictive and diagnostic values; these indices may help diagnose and treat preterm infants with BPD. 

-- The markers may be used to predict infants developing BPD, and thus help the neonatology team change management to prevent further lung trauma at 3 weeks, however, these markers have no value in treatment of already established BPD

2. At 6 weeks of PMA, the AUC of NLR for BPD diagnosis in premature infants was 0.693, with an optimal cutoff value of 1.1125, a sensitivity of 54% and a specificity of 83.3%. Taken together, these findings suggest that NLR can predict BPD occurrence at 3 weeks after birth and help diagnose BPD at 6 weeks after PMA, with some level of specificity--- I believe the authors are trying to say "36 weeks PMA" Please correct this.

Comments on the Quality of English Language

The English language is fine. However, there are still some minor grammatical errors. Please review the manuscript again once again grammatical errors.

Author Response

Dear Editor and reviewer:

Thank you very much for giving us an opportunity to revise our paper. Based on your valuable suggestions and reviewers’ comments, we have revised our original manuscript (article ID: 2748456). All amendments are highlighted in the manuscript and a list of answers is included in this letter.

I hope you give us a positive response for the publication of this paper.

Thank you for taking time for consideration.

Dr. Zhaofang Tian

Comments 1: Conclusion:  At 3 weeks after birth and 36 weeks of PMA, some systemic inflammation indices (like N, NLR, SIRI) of preterm infants with BPD have certain predictive and diagnostic values; these indices may help diagnose and treat preterm infants with BPD. 

-- The markers may be used to predict infants developing BPD, and thus help the neonatology team change management to prevent further lung trauma at 3 weeks, however, these markers have no value in treatment of already established BPD

Response 1: This is a very good opinion.We have changed in the article using green markers.

Comments 2: At 6 weeks of PMA, the AUC of NLR for BPD diagnosis in premature infants was 0.693, with an optimal cutoff value of 1.1125, a sensitivity of 54% and a specificity of 83.3%. Taken together, these findings suggest that NLR can predict BPD occurrence at 3 weeks after birth and help diagnose BPD at 6 weeks after PMA, with some level of specificity--- I believe the authors are trying to say "36 weeks PMA" Please correct this.

Response 2: Thank you for pointing out our mistakes.We have revised this.

Reviewer 2 Report (New Reviewer)

Comments and Suggestions for Authors

The authors present a revision of their study evaluating the influence of markers of inflammation on the development of BPD. The revision has added some important changes/modifications and represents a nice study with some relatively novel but important data. That said, while I did not have opportunity to review the original submission, I have several concerns about the revision as outlined below.

-          Introduction:

o   Pg 1, reference 2 – it would be helpful to add the details of the population in which the incidence of BPD was found to be 29.8%  - all NICU patients, only premature infants, only premature below a certain gestational age? Please add this clarification

o   Pg 2, para 2 – I would suggest changing the word “cheap” to “inexpensive”. Same sentence, can you explain what is meant by the phrase “easily accepted by the public”? Not sure what the relevance is here

-          Methods:

o   Pg 2, “study subjects” – The exclusion criteria read a bit awkwardly. 1st, the abbreviation PMA needs to be spelled out before being as an abbreviation. 2nd, the sentence describing that patients must remain hospitalized at 36 weeks PMA is awkward – consider simply stating it as “2: remain hospitalized beyond 36 weeks PMA”. Same goes for the sentence excluding subjects who were discharged before reaching a PMA age of 36 weeks – reword it please.

o   For exclusion criteria, it is implied earlier in the section that infants who were hospitalized for 21 days or less were excluded but this is not included in the formal listing of exclusion criteria – please reconcile

o   Given some of the comments about inclusions and exclusions from the multivariate analysis, please provide a more detailed explanation about how the multivariate analysis was derived and performed. There appear to be inconsistencies from conventional ways of performing this analysis.

-          Results:

o   Pg 3, univariate analysis – I think I know what is meant but maybe it would be worth clarifying the term “multiple  births – I presume it means twins, triplet (or more) but some might interpret it as being a non-first born child.

o   Table 1 – please define what is meant by “congenital pneumonia” Were these children born with or developed soon after birth a bacterial pneumonia or do you mean a prenatally acquired pneumonia (usually viral)?

o   There are factors other than the 7 included in the multivariate analysis that were significant in the univariate analysis (eg natal dexamethasone, EOS) that were not included in the multivariate analysis. Similarly, hood oxygen for 5-7 days was NOT significant in univariate analysis, which normally excludes a criterion from inclusion in the multivariate analysis. Please explain the rationale behind this. Along these lines, why was NLR the only marker of inflammation included in the multivariate analysis as several of these were significant in univariate analysis.  Please explain the rationale behind this.

o   I worry about presenting much data on differences between the 2 BPD categories due to the small sample size of the study. I might suggest removing these references as I suspect your study did not have adequate power to really make any analysis here.  

-          Discussion:

o   Pg 7, last line – per above questions about the way the multivariate analysis was performed, please explain the inclusion of hood O2 (5-7 l) as a significant association with BPD development.

o   Pg 8, para 2 – I think this paragraph is important but does not really incorporate the findings of the current study – I suggest either significantly abbreviating it (as the role of inflammation in BPD development has been well described) OR use it to highlight the significance of the current study’s findings.

o   It is well accepted that a significant physiologic stress (critical illness included) causes neutrophil demargination, which would alter the NLR ratio. I don’t see how this phenomenon, which is always a consequence of acute inflammation (or not necessarily a primary cause of inflammation), is discussed in the light of the current study’s interest in using the NLR as a predictive tool. Please add comment here.

o   Pg 9, para 1 – it is indicated that the current study and that if Sun differ, perhaps in part due to differences in BPD diagnostic criteria. Perhaps a sentence or 2 describing the major differences between these 2 iterations, would be in order.

o   Pg 9, para 2 – I appreciate the clarification regarding the use of glucocorticoids and why their use was higher in the BPD group. However, as steroids are generally used as ANTI-inflammatory drugs, a high percentage of infants who received steroids appear to have developed BPD (data not included but inferred from the presented data). Steroids have also been associated with an immediate, post-dosing increase in serum neutrophils. I think it would be appropriate to have a bit more discussion about this finding, even if it is purely speculative. Similarly, while the presence of increased markers of inflammation appear to predict the development of BPD, there is minimal discussion of how these findings might be incorporated into therapies designed to mitigate the development of or severity of BPD and this absence is unfortunate as there are several references throughout the text to treatment of BPD.

o   There is no discussion of the limitations of the current study which must be included.

Comments on the Quality of English Language

The manuscript will require only minor copyediting for grammar and diction.

Author Response

Dear Editor and reviewer:

Thank you very much for giving us an opportunity to revise our paper. Based on your valuable suggestions and reviewers’ comments, we have revised our original manuscript (article ID: 2748456). All amendments are highlighted in the manuscript and a list of answers is included in this letter.

I hope you give us a positive response for the publication of this paper.

Thank you for taking time for consideration.

Dr. Zhaofang Tian

Comments 1: Introduction:

o   Pg 1, reference 2 – it would be helpful to add the details of the population in which the incidence of BPD was found to be 29.8%  - all NICU patients, only premature infants, only premature below a certain gestational age? Please add this clarification

o   Pg 2, para 2 – I would suggest changing the word “cheap” to “inexpensive”. Same sentence, can you explain what is meant by the phrase “easily accepted by the public”? Not sure what the relevance is here

Response 1: Thank you for your opinion.

  • We have added it --“premature infants with gestational age<32 weeks”.
  • We have changedthe word “cheap” to “inexpensive”.We have deleted “easily accepted by the public”.

Comments 2:Methods:

o   Pg 2, “study subjects” – The exclusion criteria read a bit awkwardly. 1st, the abbreviation PMA needs to be spelled out before being as an abbreviation. 2nd, the sentence describing that patients must remain hospitalized at 36 weeks PMA is awkward – consider simply stating it as “2: remain hospitalized beyond 36 weeks PMA”. Same goes for the sentence excluding subjects who were discharged before reaching a PMA age of 36 weeks – reword it please.

o   For exclusion criteria, it is implied earlier in the section that infants who were hospitalized for 21 days or less were excluded but this is not included in the formal listing of exclusion criteria – please reconcile

o   Given some of the comments about inclusions and exclusions from the multivariate analysis, please provide a more detailed explanation about how the multivariate analysis was derived and performed. There appear to be inconsistencies from conventional ways of performing this analysis.

Response 2:This is a very good opinion.

  • We have reworded it.
  • We corrected it--“ those who were discharged before reaching 36 weeks of PMA and those with a length of hospital stay of <21 days”.
  • We chose factors that were significant in the univariate analysis. A multivariatelogistic regression model was adjusted for birth weight, SGA, days of O2 therapy, O2 inhalation in the box (5 L/min), days of CPAP therapy, mechanical ventilation time and N, NLR, SIRI at 3weeks.

Comments 3: Results:

o   Pg 3, univariate analysis – I think I know what is meant but maybe it would be worth clarifying the term “multiple  births – I presume it means twins, triplet (or more) but some might interpret it as being a non-first born child.

o   Table 1 – please define what is meant by “congenital pneumonia” Were these children born with or developed soon after birth a bacterial pneumonia or do you mean a prenatally acquired pneumonia (usually viral)?

o   There are factors other than the 7 included in the multivariate analysis that were significant in the univariate analysis (eg natal dexamethasone, EOS) that were not included in the multivariate analysis. Similarly, hood oxygen for 5-7 days was NOT significant in univariate analysis, which normally excludes a criterion from inclusion in the multivariate analysis. Please explain the rationale behind this. Along these lines, why was NLR the only marker of inflammation included in the multivariate analysis as several of these were significant in univariate analysis.  Please explain the rationale behind this.

o   I worry about presenting much data on differences between the 2 BPD categories due to the small sample size of the study. I might suggest removing these references as I suspect your study did not have adequate power to really make any analysis here.

Response 3: Thank you for your good advice on our research.

  • We have changed “univariate analysis”to ”baseline characteristics” and changed “multiple births” to” multiple pregnancies”.
  • “Congenital pneumonia”means “ prenatally acquired pneumonia”.
  • Following your suggestion, we re-performed the multivariate analysis, hood oxygen for 5-7 days was excluded, and N, SIRI were included. Because many factors were significant in the univariate analysis, and postnatal dexamethasome treatment and EOS were not reported as an independent risk factor for BPD in previous studies,we excluded them. we can consider to include them in the future study.
  • We have removed it.

Comments 4: Discussion:

o   Pg 7, last line – per above questions about the way the multivariate analysis was performed, please explain the inclusion of hood O2 (5-7 l) as a significant association with BPD development.

o   Pg 8, para 2 – I think this paragraph is important but does not really incorporate the findings of the current study – I suggest either significantly abbreviating it (as the role of inflammation in BPD development has been well described) OR use it to highlight the significance of the current study’s findings.

o   It is well accepted that a significant physiologic stress (critical illness included) causes neutrophil demargination, which would alter the NLR ratio. I don’t see how this phenomenon, which is always a consequence of acute inflammation (or not necessarily a primary cause of inflammation), is discussed in the light of the current study’s interest in using the NLR as a predictive tool. Please add comment here.

o   Pg 9, para 1 – it is indicated that the current study and that if Sun differ, perhaps in part due to differences in BPD diagnostic criteria. Perhaps a sentence or 2 describing the major differences between these 2 iterations, would be in order.

o   Pg 9, para 2 – I appreciate the clarification regarding the use of glucocorticoids and why their use was higher in the BPD group. However, as steroids are generally used as ANTI-inflammatory drugs, a high percentage of infants who received steroids appear to have developed BPD (data not included but inferred from the presented data). Steroids have also been associated with an immediate, post-dosing increase in serum neutrophils. I think it would be appropriate to have a bit more discussion about this finding, even if it is purely speculative. Similarly, while the presence of increased markers of inflammation appear to predict the development of BPD, there is minimal discussion of how these findings might be incorporated into therapies designed to mitigate the development of or severity of BPD and this absence is unfortunate as there are several references throughout the text to treatment of BPD.

o   There is no discussion of the limitations of the current study which must be included.

Response 4: Thank you for your very good comments.

  • We have re-corrected this.
  • We have changed in Pg 8, para 2 using green markers.
  • We believe that with increasing NLR, promoting inflammatory effects, leading to BPD. We can predict the occurrence of BPD by the increase of NLR at 3 weeks.
  • We added it in the article--These two are different in the inclusion population, assessment nodes and classification.
  • We have added the discussion in Pg 9,para 2-3.
  • Our limitation isthe small number of EOS cases and the higher use of postnatal steroids. And this paper is a single-center clinical study with a small sample size. Additional multicenter, large-scale clinical investigations are necessary.

This manuscript is a resubmission of an earlier submission. The following is a list of the peer review reports and author responses from that submission.

Round 1

Reviewer 1 Report

Comments and Suggestions for Authors

This is an interesting article and can be of some interest to the readers. The study has limitations, being an observational study, done on one site and with modest sample size. The median gestational age and birth weight in the study group as well as use of postnatal steroids in the study population are much higher than current practice. There is unusually higher incidence of congenital pneumonia and neonatal sepsis within the study group. It is therefore difficult to interpret the results considering higher incidence of sepsis and postnatal steroid use. 

The authors should quote the incidence of BPD at the study site as it appears that the incidence may be higher than quoted in other studies.

The authors conclude significant higher predictive value of NLR at 3 and 36 weeks of post menstrual age in infants with BPD. It is unclear whether this is due to increased incidence of sepsis and use of postnatal steroids in this group.

The authors have repeated 3 references in the bibliography. Reference numbers 2, 11 and 18 are same. Reference 3 and 13 are same. The authors need to review references quoted in bibliography to make sure there is no repetition of references to increase the number of references.

Overall, although this is an interesting article with an interesting conclusion, there is no explanation how this will be help in the future management of BPD.

Author Response

Dear Editor and reviewer:

Thank you very much for giving us an opportunity to revise our paper. Based on your valuable suggestions and reviewers’ comments, we have revised our original manuscript (article ID: 2603878). All amendments are highlighted in the manuscript and a list of answers is included in this letter.

I hope you give us a positive response for the publication of this paper.

Thank you for taking time for consideration.

Dr. Zhaofang Tian

Comments 1:This is an interesting article and can be of some interest to the readers. The study has limitations, being an observational study, done on one site and with modest sample size. The median gestational age and birth weight in the study group as well as use of postnatal steroids in the study population are much higher than current practice. There is unusually higher incidence of congenital pneumonia and neonatal sepsis within the study group. It is therefore difficult to interpret the results considering higher incidence of sepsis and postnatal steroid use. 

Response: Thank you for your positive comments on our research. Higher gestational age and weight might related to the level of NICU  where the researcher is located. Some very premature infants were  transferred to the other NICU or died within two weeks after birth. According to the 2018 diagnostic criteria of BPD , these cases can’t be diagnosed as BPD IIIa., possibly the absence of these cases affected the results; In the study population,the higher use of postnatal steroids was related to higher mechanical ventilation rate in this group.We treated these neonates who were still mechanically ventilating one week after birth with low dose of glucocorticoid to prevent BPD and promote extubation.However, the higher incidence of neonatal sepsis might be related to the longer hospitalization time for BPD and the susceptibility to nosocomial infection. These factors will really affect the interpretation of he results in this study,and this may be the limitations of of our single-center research. which is explained and has been marked in red in the discussion section-page 9, paragraph 2, and line 9.

Comments 2: The authors should quote the incidence of BPD at the study site as it appears that the incidence may be higher than quoted in other studies.

Response: We have made the changes in the “Introduction” section-page 1, paragraph 1, and line 3.

Comments 3: The authors conclude significant higher predictive value of NLR at 3 and 36 weeks of post menstrual age in infants with BPD. It is unclear whether this is due to increased incidence of sepsis and use of postnatal steroids in this group.

Response: This is indeed a very good topic.We have explained the reasons of higher use of postnatal steroids and incidence of neonatal sepsis, which may be a common problem in the clinical management of BPD patients, and there is fewer relevant literature report at present,which is explained and has been marked in red in the discussion section-page 9, paragraph 2, and line 9. This will be the direction of our further research.

Comments 4: The authors have repeated 3 references in the bibliography. Reference numbers 2, 11 and 18 are same. Reference 3 and 13 are same. The authors need to review references quoted in bibliography to make sure there is no repetition of references to increase the number of references.

Response: Thank you for pointing out our mistakes.We have revised the references in the text-reference 4 and reference 15.

Comments 5: Overall, although this is an interesting article with an interesting conclusion, there is no explanation how this will be help in the future management of BPD.

Response:This is a very good opinion. According to the existing diagnostic criteria, the diagnosis of BPD can only be made after 4 weeks of birth. The study of these predictive markers of BPD may help us to intervene in high-risk patients with BPD as soon as possible. We will add this part to the discussion-page 9, paragraph 2, line 5.

Reviewer 2 Report

Comments and Suggestions for Authors

The authors have described an interesting prospective study to evaluate the use of inflammatory markers in prediction and diagnosis of BPD.

The following needs to be addressed before a decision regarding the paper's acceptability can be made.

1. Abstract:

- The background describes the aim but no introduction regarding the scientific rationale for the study. This needs to be corrected.

2. Introduction

- Provide reference for 'Prevalence of BPD has increased'

3. Material and methods:

- >32 weeks is not an exclusion criteria as your inclusion criteria already states infants <32 weeks GA. Please delete this

-Statistical analysis: While it is good to see that multivariate analysis showed that birthweight, SGA status, multiple gestations and days on O2 therapy were independently associated with BPD, the authors should adjust the systemic inflammatory markers for the above confounding factors using multivariate regression analysis to study their effect on BPD!

4. Results:

N, NLR, MLR and SIRI were all found to be higher in infants with BPD compared to those without BPD at 3 weeks of life and 36 weeks PMA. Can the authors explain as to why only the trends of NLR and the ROC analysis of NLR was used to predict (3 weeks) and diagnose (36 weeks PMA) BPD? Can the authors provide the ROC analyses for N, MLR and SIRI as well?

5. Discussion:

- Please elaborate and discuss the differences observed between this study and the recently published study by Cakir et al (ref 8 in the manuscript). In the previously published study, there was no difference noted in any inflammatory markers except SIRI at 36 weeks of gestation. Can the authors comment on this in discussion.

- Please list the strengths and the limitations of the study.

Comments on the Quality of English Language

The manuscript has several grammatical errors. Please revise the manuscript for English using either grammar editing software or with the help of someone fluent in English. 

Author Response

Comments 1:Abstract:

- The background describes the aim but no introduction regarding the scientific rationale for the study. This needs to be corrected.

Response 1: Thank you for pointing this out. We have made the changes in the “Abstract-Background” section.

Comments 2: Introduction

-Provide reference for 'Prevalence of BPD has increased'.

Response 2: Agree. We have made the changes in the “Introduction” section-page 1, paragraph 1, and line 3.

Comments 3: Material and methods:

- >32 weeks is not an exclusion criteria as your inclusion criteria already states infants <32 weeks GA. Please delete this

-Statistical analysis: While it is good to see that multivariate analysis showed that birthweight, SGA status, multiple gestations and days on O2 therapy were independently associated with BPD, the authors should adjust the systemic inflammatory markers for the above confounding factors using multivariate regression analysis to study their effect on BPD!

Response 3: Agree. 

-We have deleted this in the paper.

-We have made the changes in the “Results-Multivariate logistic regression analysis” and “Table 2” sections.

Comments 4: Results:

-N, NLR, MLR and SIRI were all found to be higher in infants with BPD compared to those without BPD at 3 weeks of life and 36 weeks PMA. Can the authors explain as to why only the trends of NLR and the ROC analysis of NLR was used to predict (3 weeks) and diagnose (36 weeks PMA) BPD? Can the authors provide the ROC analyses for N, MLR and SIRI as well?

Response 4: Since ROC analysis at 3 weeks and 36 weeks of PMA showed higher AUC values of NLR than N, MR and SIRI, only NLR was chosen.Next we provide the ROC analyses for N, MLR and SIRI at 3weeks and 36 weeks PMA.

Finally, we have taken your advice and checked the entire text, mainly to correct the problem of grammatical errors.

Comments 5: Discussion: 

- Please elaborate and discuss the differences observed between this study and the recently published study by Cakir et al (ref 8 in the manuscript). In the previously published study, there was no difference noted in any inflammatory markers except SIRI at 36 weeks of gestation. Can the authors comment on this in discussion.

- Please list the strengths and the limitations of the study.

Response 5: Thank you for pointing this out.

-Cakir et al. reported systemic inflammatory indices in bronchopulmonary dysplasia at birth and 36 weeks of PMA, and the results showed a difference only with SIRI, which is inconsistent with our findings. First, the BPD diagnostic criteria of 2018 were used in our study, while they used the diagnostic criteria of 2001, and there were differences in the diagnosis of BPD; Second, their study was divided into group1 (non or mild BPD) and group2 (moderate/severe BPD), comparing the two groups, while our study compared BPD with non-BPD, which may be the reason why our results differed from theirs. We have added this and has been marked in red in the “Discussion” section on page 9, paragraph 1, and line 13.

-Our strengths are the adoption of the latest diagnostic criteria, comparing multiple inflammatory markers at different time periods, and finding the significant value of NLR at 3 weeks and 36 weeks of PMA.

Our limitation is that this paper is a single-center clinical study with a small sample size. Additional multicenter, large-scale clinical investigations are necessary.

Reviewer 3 Report

Comments and Suggestions for Authors

This is a very interesting article. In this paper the authors try to identify and Predictive and Diagnostic Values of Systemic Inflammatory Indices in Bronchopulmonary Dysplasia.

The paper is well written but it has some statistical issues:

1. Tables and graph are no autoexplained. the tables must shows witch central measure and dispersión measures are shown.

this problem is enhanced in table 4, where there are not dispersion values of the inflammatory indices.

2. Figura 1. the variables are cuantitative ( not frequencies ) so the appropiate graph must be a box plot and not a histogram of frequencies.

3. ROC curve of figure 2  is not clear. In order to improve the comprenhension

the authors can show the values of AUC and the 95%CI at different weeks, also the best Younden test.

The different systemic inflammatory indices need to be adjusted (in a multiple logistic regression model for each week ) for the main confusing  values just like SMGA, birth weigh and/or multiple pregnancies .

With the results with very low prediction and low performance of the different  inflammatory indices, the conclusions (we observed that some systemic inflammatory indices have certain clinical value in predicting and diagnosing BPD at 3 weeks) seems to loose validity.

Author Response

Dear Editor and reviewer:

Thank you very much for giving us an opportunity to revise our paper. Based on your valuable suggestions and reviewers’ comments, we have revised our original manuscript (article ID: 2603878). All amendments are highlighted in the manuscript and a list of answers is included in this letter.

I hope you give us a positive response for the publication of this paper.

Thank you for taking time for consideration.

Dr. Zhaofang Tian

Comments 1:Tables and graph are no autoexplained. the tables must shows witch central measure and dispersión measures are shown.this problem is enhanced in table 4, where there are not dispersion values of the inflammatory indices.

Response 1: Thank you for pointing this out. We have made the changes under every table and graph.

Comments 2: Figura 1. the variables are cuantitative ( not frequencies ) so the appropiate graph must be a box plot and not a histogram of frequencies.

Response 2: Agree. We have changed in Figure 1.

Comments 3: ROC curve of figure 2  is not clear. In order to improve the comprenhension.the authors can show the values of AUC and the 95%CI at different weeks, also the best Younden test.

Response 3: Thank you for pointing this out. We have added table 5.

Comments 4: The different systemic inflammatory indices need to be adjusted (in a multiple logistic regression model for each week ) for the main confusing  values just like SMGA, birth weigh and/or multiple pregnancies .

Response 4: Agree. We have made the changes in the “Results-Multivariate logistic regression analysis” and “Table 2” sections.

Comments 5: With the results with very low prediction and low performance of the different  inflammatory indices, the conclusions (we observed that some systemic inflammatory indices have certain clinical value in predicting and diagnosing BPD at 3 weeks) seems to loose validity.

Response 5:Thank you for pointing this out. We have changed the conclusions that we observed that some systemic inflammatory indices (like N、NLR、MLR and SIRI) have certain clinical value in predicting BPD at 3 weeks and diagnosing BPD after birth and 36 weeks of PMA.